# Dietary *n*-6/*n*-3 Ratio Influences Brain Fatty Acid Composition in Adult Rats

**DOI:** 10.3390/nu12061847

**Published:** 2020-06-21

**Authors:** Thomas Horman, Maria F. Fernandes, Maria C. Tache, Barbora Hucik, David M. Mutch, Francesco Leri

**Affiliations:** 1Department of Psychology and Neuroscience, University of Guelph, Guelph, ON N1G 2W1, Canada; thorman@uoguelph.ca (T.H.); mariafer@uoguelph.ca (M.F.F.); 2Department of Human Health and Nutritional Sciences, University of Guelph, Guelph, ON N1G 2W1, Canada; mariacristinatache@gmail.com (M.C.T.); bhucik@uoguelph.ca (B.H.); dmutch@uoguelph.ca (D.M.M.)

**Keywords:** polyunsaturated fatty acids, linoleic acid, α-linolenic acid, diet, rat, brain

## Abstract

There is mounting evidence that diets supplemented with polyunsaturated fatty acids (PUFA) can impact brain biology and functions. This study investigated whether moderately high-fat diets differing in *n*-6/*n*-3 fatty acid ratio could impact fatty acid composition in regions of the brain linked to various psychopathologies. Adult male Sprague Dawley rats consumed isocaloric diets (35% kcal from fat) containing different ratios of linoleic acid (*n*-6) and alpha-linolenic acid (*n*-3) for 2 months. It was found that the profiles of PUFA in the prefrontal cortex, hippocampus, and hypothalamus reflected the fatty acid composition of the diet. In addition, region-specific changes in saturated fatty acids and monounsaturated fatty acids were detected in the hypothalamus, but not in the hippocampus or prefrontal cortex. This study in adult rats demonstrates that fatty acid remodeling in the brain by diet can occur within months and provides additional evidence for the suggestion that diet could impact mental health.

## 1. Introduction

There are important links between diet and psychopathology [1]. For instance, diets characterized by high *n*-6/*n*-3 poly-unsaturated fatty acid (PUFA) ratios have been linked to a variety of affective, cognitive and behavioral deficits [2,3]. Moreover, patients suffering from conditions ranging from Alzheimer’s to major depression display lower *n*-3 PUFA and higher *n*-6 PUFA levels in brain and plasma [4,5]. PUFA are linked to mental health because of the many functions they regulate, including neurotransmission, synaptic function, neuronal survival, and neuroinflammation [3,6]. For example, diets with a high *n*-6/*n*-3 ratio can reduce monoamine neurotransmission, impair neurogenesis, alter hypothalamic-pituitary-adrenal (HPA) stress axis activity, and increase depression-like behavior [7,8,9,10,11]. Conversely, diets with low *n*-6/*n*-3 ratios are associated with cognitive benefits and reduced depression risk [12,13,14]. 

Linoleic acid (LA; 18:2*n*-6) and α-linolenic acid (ALA; 18:3*n*-3) are essential fatty acids that cannot be synthesized de novo and must therefore be consumed in the diet. Following their ingestion, LA and ALA can be converted into other important fatty acids, including arachidonic acid (AA; 20:4n-6) and docosahexaenoic acid (DHA; 22:6*n*-3), which are the predominant long-chain PUFAs (LC-PUFA) in the brain. While it is recognized that the typical Western diet is unbalanced with a high *n*-6/*n*-3 ratio, there is still some debate regarding the optimal composition of PUFA in the diet [15]. However, most studies examining the role of dietary PUFA on brain biology have supplemented with eicosapentaenoic acid (EPA; 20:5*n*-3) and DHA to improve the *n*-6/*n*-3 ratio in the diet [14]. Comparatively less is known about the effects of LA and ALA, which have their own distinct bioactivity compared to LC-PUFA. Importantly, LA and ALA compete for the same desaturase and elongase enzymes, which means that their rate of conversion into AA and DHA depends on relative amounts consumed [16]. In addition, the brain contains the second highest concentration of total lipid in the body, after adipose tissue, and it is sensitive to changes in dietary fat intake, but in a region-specific manner [17]. For example, mice fed a diet containing a balanced 7:1 ratio of LA:ALA displayed higher PUFA levels in the prefrontal cortex (PFC), hippocampus (HPC) and hypothalamus (HYP), compared to the rest of the cortex, the cerebellum and brainstem, suggesting that these regions may be more sensitive to changes in dietary fats [17].

Therefore, the objective of this study was to establish whether high and low *n*-6/*n*-3 (LA:ALA) ratio diets could alter fatty acid content in different regions of the rat brain. Adult male Sprague Dawley rats were selected because of their widespread use in animal models of human psychopathology [18]. The diets employed in this study were selected to explore the impact of different LA:ALA ratios using isocaloric diets, rather than supplementation studies that are more commonly used, in order to capture the biological interactions of these essential fats [19,20,21]. The focus was on fatty acid content in the PFC, HPC, and HYP because of the high abundance of PUFA in these regions [17] and because altered fatty acid composition in these regions is linked to prevalent and debilitating psychopathologies such as substance dependence, obsessive compulsive disorder, major depression, Alzheimer’s, and metabolic disfunctions [3,14]. 

## 2. Materials and Methods 

### 2.1. Animals and Diets

Adult male Sprague Dawley rats (6–7 weeks old) were received from Charles River (St-Constant, QC, Canada). All rats were housed in environmentally controlled rooms (22–24 °C) with standard environmental enrichment and ad libitum access to water and diets for 2 months. Diets were isocaloric, with macronutrient composition consisting of 35% kcal from fat, 20% kcal from protein and 45% kcal from carbohydrates (Table 1; diets from Research Diets, Inc., New Brunswick, NJ, USA). Diets differed only in regard to the source of dietary oil. Rats were randomly assigned to one of the following diet groups: (1) high *n*-6/*n*-3 ratio (~600:1) diet containing safflower oil (SD; D15010801) and (2) low *n*-6/*n*-3 ratio (~0.25:1) diet containing flaxseed oil (FD; D15010802). Body weight and diet intake were assessed daily at the onset of the dark period. All procedures were approved by the Animal Care Committee of the University of Guelph and were carried out in accordance with the recommendations of the Canadian Council on Animal Care (Animal Utilization Protocol # 3609; Approved Timeline: Oct 2016–Oct 2020). 

### 2.2. Prefrontal Cortex, Hippocampus, and Hypothalamus Dissections

Rats were sacrificed with carbon dioxide and the brains were rapidly collected and flash frozen (isopentane bath maintained between −30 °C and −20 °C) and stored at −80 °C until micro-dissection. Frozen brains were sliced into coronal sections using a rat brain matrix and mounted onto slides maintained on dry ice. Nuclei were micro-dissected using brain tissue punches (Stoelting, Inc., Kiel, WI, USA), as described elsewhere [22].

### 2.3. Fatty Acid Analysis by Gas Chromatography 

All solvents and reagents were obtained from Fisher Scientific (Toronto, ON, Canada). Prefrontal cortex (PFC), hippocampal (HPC) and hypothalamic (HYP) samples (0.1 g) were maintained on ice and homogenized in a 0.1 M KCl solution. Total fatty acid methyl esters were extracted from samples with a chloroform:methanol (2:1, *v*/*v*) solution, according to Folch et al. [23]. Complete details of the fatty acid analysis have been previously published [24]. Fatty acid methyl esters were detected using an Agilent 6890A gas chromatograph with flame ionization detector (Agilent Technologies, Palo Alto, CA, USA) and separated on DBFFAP fused silica capillary column (15 m, 0.1 µm film thickness, 0.1 mm i.d.; Agilent Technologies, USA). Fatty acid peaks were identified by comparison to retention times of known fatty acid standard peaks (Nu-Chek-Prep, Elysian, MN, USA) using EZChrom Elite software (Version 3.3.2) to determine the relative abundance of individual fatty acids. Fatty acids are reported as a percentage of total fatty acids detected. The n-3 PUFA:LC-PUFA ratio was calculated as the sum of EPA+DPAn-3+DHA/sum of all PUFA ≥ 20 carbons.

### 2.4. Statistical Analyses

Data were analyzed by *t*-test, or Mann–Whitney Rank Sum test for data not normally distributed, within individual brain regions (Sigma Plot, Version 12). Data are reported as mean ± SEM, and statistical significance was considered at *p* < 0.05, corrected using the Bonferroni method for multiple comparisons.

## 3. Results

After consuming a high n-6/n-3 ratio (~600:1) diet containing safflower oil (SD) or a low n-6/n-3 ratio (~0.25:1) diet containing flaxseed oil (FD) for two months, no differences in body weight (g; FD = 558.87 ± 17.77; SD = 570.67 ± 13.74) or caloric intake (kcal/100 g; FD = 5.88 ± 0.13; SD = 5.93 ± 0.06) were detected between the two diet groups prior to tissue collection.

The relative fatty acid composition of the PFC, HPC, and HYP of rats fed the high and low n-6/n-3 ratio diets are presented in Table 2. In each diet group, total saturated fatty acids (SFA) represented the highest percentage of fatty acids (PFC ~49%; HPC ~46%; HYP ~44%), followed by monounsaturated fatty acids (MUFA, PFC ~21%; HPC ~26%; HYP ~30%), n-6 PUFA (PFC ~15%; HPC ~15%; HYP ~15%), and n-3 PUFA (PFC ~13%; HPC ~12%; HYP ~12%). The SD group had significantly higher levels of total SFA (+7.8%, *p* < 0.0001; Figure 1a), as well as significantly lower levels of total MUFA (−15.8%, *p* < 0.001; Figure 1b) in the HYP only. In each brain region, total n-3 and n-6 PUFA composition reflected that of the diets. That is, total n-3 PUFA was higher in FD compared to SD animals (PFC +21.4%, *p* < 0.001; HPC +26.2%, *p* < 0.01; HYP +22.1%, *p* < 0.001; Figure 1c), while total n-6 PUFA was higher in SD compared to FD animals (PFC +17.6%, *p* < 0.001; HPC +23.1%, *p* < 0.0001; HYP +26.4, *p* < 0.0001; Figure 1d). These differences were also reflected in the n-3 PUFA:LC-PUFA ratio; (Figure 1e) and the DHA:AA ratio (Figure 1f).

Differences in the relative proportions of individual fatty acids were also observed between the diet groups (Table 2). Regarding SFA and MUFA in the HYP, the SD group exhibited significantly higher levels of 16:0 and 18:0 compared to the FD group, and significantly lower levels of 20:0, 18:1*n*-9, and 20:1*n*-9. In the PFC, the SD group exhibited significantly higher levels of 18:1*n*-7. PUFA expression was consistent across all three brain regions. Specifically, linoleic acid (LA), arachidonic acid (AA), and 22:4*n*-6 exhibited higher levels in the SD group compared to the FD group in all three brain regions. The SD group also showed significantly higher levels of 22:2*n*-6 compared to the FD group in only the HPC; however; similar trends were seen in the PFC and HYP. In the PFC, docosahexaenoic acid (DHA) exhibited significantly lower levels in the SD group compared to the FD group; however, similar trends were detected in the HPC and HYP. ALA and EPA were only detected in all 3 brain regions at low levels in only rats fed the FD.

## 4. Discussion

Diets that differ in their relative fatty acid composition are purported to influence brain structure and function. The present study found that feeding adult rats a diet with a high *n*-6/*n*-3 ratio (~600:1) or a low *n*-6/*n*-3 ratio (~0.25:1) for two months altered the PUFA profile in the PFC, HPC, and HYP in a manner that reflected the fatty acid composition of their diet. These diets also altered SFA and MUFA levels in the HYP, but not in the PFC and HPC. Overall, these results indicate that: (1) fatty acid remodeling in the adult rat brain can occur within months, (2) dietary ALA is capable of inducing this remodeling, and (3) that the HYP may be particularly sensitive in comparison to the PFC and HPC.

Our analyses revealed that PUFA levels in all brain regions analyzed reflected dietary fat composition: higher total and individual n-6 PUFA in SD-fed rats and higher total and individual n-3 PUFA in FD-fed rats. The observation in SD-fed rats is consistent with previous work investigating the influence of n-3 PUFA deficient diets on brain PUFA composition in mice [17,25]. Joffre et al. investigated the effect of long term (4 months) consumption of an n-3 PUFA deficient (*n*-6/*n*-3 ratio of >500:1) or a more balanced (*n*-6/*n*-3 ratio of 6.7:1) diet in the cortex, HPC, HYP, PFC, brainstem, and cerebellum of C57 mice [17]. While some regional variability was observed, mice fed *n*-3 PUFA deficient diets exhibited lower levels of total *n*-3 PUFA and DHA in all regions investigated. In contrast, relative *n*-6 PUFA levels (including AA) were increased in these same mice, ultimately leading to a general increase in AA/DHA ratio in the various brain regions. There are some differences between the present study and the important work by Joffre et al. that are noteworthy to highlight. Briefly, Joffre et al. examined the effect of different n-3 PUFA diets in various mouse models, reporting that PUFA levels varied based on dietary manipulation, as well as strain and age of mice [17]. The current study complements the Joffre et al. findings by reporting the effect of essential dietary fat ratios on brain FA levels in rats. Further, our goal was to examine the effects of moderate fat diets (35% kcal/d) on various regions of the brain, while Joffre et al. used low-fat diets (~11% kcal/d, as calculated from their published diet table). The level of fat used in the current study was chosen to reflect levels that are typical of the North American diet. Finally, the feeding protocol used in the present study was half as long as that used by Joffre et al. (2 months vs. 4 months), which allowed us to show that fatty acid remodeling in the brain of rats can happen relatively quickly.

Despite FD-fed rats consuming high levels of ALA, this essential fatty acid was only detected at very low levels in the brain. However, FD-fed rats also had DHA levels that were >360 times higher than ALA levels in the PFC and ~60 times higher than ALA levels in the HPC and HYP. Differences between DHA and EPA content were similarly observed, which is consistent with previous work [17]. There are at least two possible explanations for the observation that the FD diet resulted in higher levels of DHA but little-to-no ALA. First, it could be that most of the ALA that enters the brain is rapidly converted to DHA. However, this seems unlikely given that EPA (which is produced from ALA to a greater extent than DHA) was generally present at levels equivalent to ALA. Rather, work by DeMar and colleagues showed that although ALA enters the brain rapidly, it is primarily oxidized and ≤0.2% is converted into DHA [26]. Interestingly, DHA levels in the PFC, HPC, and HYP were all greater in FD-fed rats compared to SD-fed rats, although statistical significance was only achieved in the PFC. Thus, the difference in DHA levels between the two groups of rats after 8 weeks was not as large as expected. However, this is perhaps not surprising given DHA’s importance for normal brain structure, function and metabolism. Using labelled fatty acid tracers, DeMar et al. showed that brain DHA levels were reduced by 37% after feeding rats an n-3 PUFA deficient diet (i.e., a safflower diet similar to that used in the current study) for 15 weeks from weaning [27]. Moreover, this reduction in DHA was unchanged over a subsequent 60-day period, indicating little additional loss in brain DHA content. Notable differences between this past study and our study included the age at which rats began to have access to the *n*-3 PUFA deficient diet and the duration of the feeding trial. In our study, rats were well past weaning when they began the experimental diet, and our study was shorter in duration. Thus, it is plausible that we would need to extend the feeding trial for longer before detecting reductions in DHA in rats fed the SD compared to FD. Second, DHA levels in the brain are sustained primarily by peripheral, rather than central, synthesis from ALA. This explanation is consistent with recent work by Lacombe and colleagues, who demonstrated that rats fed preformed DHA showed higher levels of brain DHA compared to animals fed an ALA-based diet [28]. Thus, the current study supports the idea that DHA taken up in the brain may be formed from the conversion of ALA that occurs elsewhere in the body (e.g., liver) and that mechanisms exist to preserve brain DHA levels and buffer against short-term dietary deficiencies in *n*-3 PUFA.

Interestingly, SD-fed rats exhibited higher levels of total SFA, palmitic, and stearic acid, as well as lower levels of total MUFA, oleic, and eicosenoic acid levels compared to the FD-fed rats in the HYP, but not in the HPC or PFC. Recently, Rey et al. reported no change in SFA or MUFA in the HPC of mice after 2 months of *n*-3 PUFA deficient or LC-PUFA supplemented diets [29]. In contrast, Joffre et al. reported that *n*-3 PUFA deficiency elevated total SFA and palmitic acid levels, as well as lowered total MUFA, oleic, and eicosenoic acid levels in the cortex of C57 mice after 4 months [17]. Thus, elevated SFA (specifically palmitic acid) levels found only in the rat HYP after only 2 months suggests that the HYP may be more sensitive to dietary modulation than the HPC or PFC. Unfortunately, literature investigating region-specific effects of diet on brain PUFA composition is limited and more research is required to elucidate the time-course of these changes. Having said that, given that palmitate is generally positioned as a pro-inflammatory fatty acid [30], the current results may have implications for neuroinflammation in the HYP.

Overall, these results have potential implications for understanding links between diet, brain biology, and psychopathology. Here we show that high and low *n*-6/*n*-3 (LA:ALA) ratio diets differentially alter fatty acid composition in three regions of the adult rat brain after only 2 months of feeding. Although not directly tested, it is interesting to consider possible functional ramifications of these diets. For example, the HPC and PFC play central roles in psychopathologies that involve impaired learning, memory, stress reactivity, and emotional control [31,32]. *N*-3 PUFA deficiency in these regions was reported to impair neurogenesis and alter monoamine activity required for normal function [7,8,9,10]. The HYP plays a central role in mediating metabolic function and the HPA stress axis [33,34], and *n*-3 deprivation promoted HPA-axis hyperactivity, an exaggerated stress response, and altered insulin signaling [11,24]. Importantly, dietary *n*-3 PUFA supplementation (typically consisting of EPA and/or DHA treatment) has been shown to attenuate positive and negative symptoms in patients suffering from schizophrenia [35], improve treatment efficacy in patients suffering from depression [36], and reduce cognitive decline in patients suffering from Alzheimer’s [37]. Therefore, these current data in rats bolster the evidence demonstrating that essential dietary fats can alter fatty acid composition in the brain [13,14,38] and show that consumption of a moderately high-fat diet rich in ALA (but deficient in EPA and DHA) can help maintain levels of brain DHA better than a moderately high-fat diet containing only trace amounts of ALA. 

## Figures and Tables

**Figure 1 nutrients-12-01847-f001:**
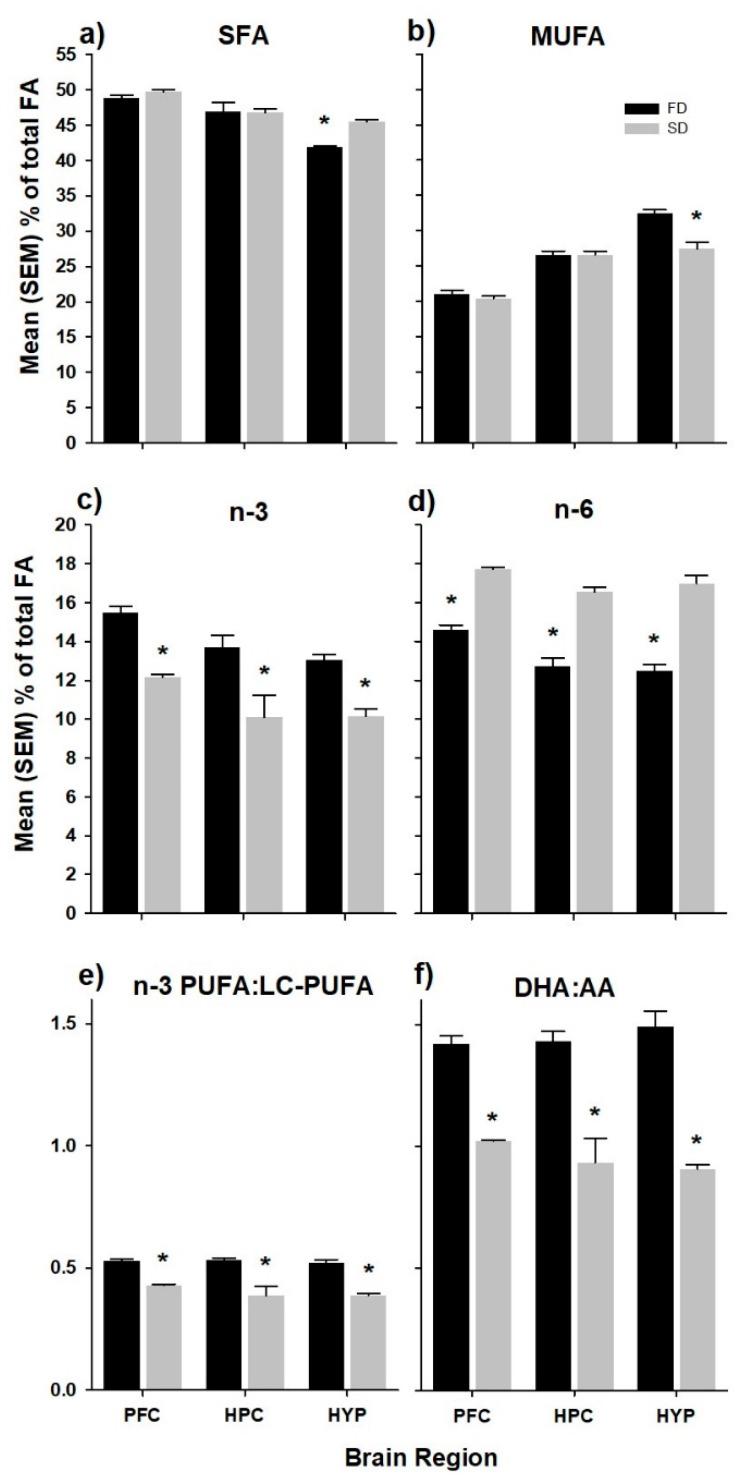
Prefrontal Cortex (PFC), hippocampus (HPC), and hypothalamus (HYP) composition of (**a**) total saturated fatty acids (SFA); (**b**) total monounsaturated fatty acids (MUFA); (**c**) total *n*-3 polyunsaturated fatty acids (PUFA); (**d**) total *n*-6 PUFA; (**e**) *n*-3 PUFA:LC-PUFA ratio, which was calculated as the sum of EPA+DPAn-3+DHA/sum of all PUFA ≥ 20 carbons; and (**f**) DHA:AA ratio in rats fed experimental diets. Values correspond to % fatty acid ± standard error of the mean (*n* = 6–8 rats per diet group). The * denotes statistical difference between diet groups within brain regions (α ≤ 0.05). FD, flaxseed diet (low *n*-6/*n*-3 ratio); SD, safflower diet (high *n*-6/*n*-3 ratio).

**Table 1 nutrients-12-01847-t001:** Composition of experimental diets. Composition of AIN-93G modified diets as provided by manufacturer, Research Diets. Diet product numbers are D15010802 (FD) and D15010801 (SD).

	Flaxseed Diet (FD)	Safflower (SD)
	kcal (%)	kcal (%)
Protein	20	20
Carbohydrate	45	45
Fat	35	35
kcal/gm	4.47	4.47
Ingredient (g/kg diet)		
Soybean oil	0	0
Flaxseed oil	154	0
Safflower oil	0	154
t-Butylhydroquinone	0.0228	0.0028
Casein	200	200
L-Cystine	3	3
Corn Starch	208	208
Maltodextrin 10	132	132
Sucrose	100	100
Cellulose, BW200	50	50
Mineral Mix S10022G	35	35
Vitamin Mix V10037	10	10
Choline Bitartrate	2.5	2.5
Fatty Acid (g fat/kg diet)		
14:0	-	-
16:0	8.2	9.9
16:1n-7	-	-
18:0	6.3	3.5
18:1n-9	31.1	18.5
18:2n-6	19.6	120.7
18:3n-3	82	0.2
20:0	-	-
20:1n-9	-	-
22:0	-	-
24:0	-	-
*n*-6:*n*-3 ratio	~0.25	~600
Relative Composition		
% SFA	9.80%	8.80%
% MUFA	21.10%	12.10%
% n-3 PUFA	55.70%	1%
% n-6 PUFA	13.30%	79%

**Table 2 nutrients-12-01847-t002:** Prefrontal Cortex (PFC), hippocampus (HPC), and hypothalamus (HYP) fatty acid composition of rats fed experimental diets. Values correspond to % fatty acid ± standard error of the mean (*n* = 6–8 rats per diet group). The * denotes statistical difference between diet groups within brain regions (α ≤ 0.05, with Bonferroni correction for multiple comparisons 0.05/21 = 0.00238). FD, flaxseed diet; SD, safflower diet.

	PFC		Hippocampus		Hypothalamus	
Fatty Acid	FD	SD	FD	SD	FD	SD
14:0, Myristic	0.47 ± 0.03	0.51 ± 0.03	0.28 ± 0.02	0.33 ± 0.05	0.18 ± 0.01	0.21 ± 0.01
16:0, Palmitic	25.39 ± 0.50	25.92 ± 0.36	24.13 ± 1.78	22.85 ± 0.30	18.14 ± 0.13	20.67 ± 0.55 *
18:0, Stearic	21.84 ± 0.15	22.35 ± 0.21	20.45 ± 0.46	21.25 ± 0.24	20.61 ± 0.09	21.82 ± 0.07 *
20:0, Arachidic	0.22 ± 0.07	0.17 ± 0.06	0.48 ± 0.46	0.52 ± 0.02	0.73 ± 0.01	0.56 ± 0.04 *
22:0, Behenic	0.52 ± 0.16	0.39 ± 0.15	0.82 ± 0.08	0.97 ± 0.03	1.07 ± 0.02	1.14 ± 0.07
24:0, Lignoceric	0.38 ± 0.09	0.38 ± 0.01	0.79 ± 0.05	0.87 ± 0.04	1.18 ± 0.07	1.03 ± 0.16
Total SFA	48.83 ± 0.37	49.72 ± 0.33	46.95 ± 1.28	46.79 ± 0.50	41.90 ± 0.13	45.44 ± 0.32 *
16:1*n*-7, Palmitoleic	0.35 ± 0.08	0.39 ± 0.07	0.51 ± 0.07	0.43 ± 0.01	0.55 ± 0.05	0.72 ± 0.07
18:1*n*-9, Oleic	16.02 ± 0.25	15.04 ± 0.18	19.66 ± 0.35	18.79 ± 0.44	22.39 ± 0.29	18.64 ± 0.64 *
18:1n-7, Vaccenic	3.14 ± 0.06	3.35 ± 0.02 *	3.57 ± 0.12	3.8 ± 0.05	4.20 ± 0.08	3.95 ± 0.10
20:1n-9, Eicosenoic	0.92 ± 0.11	0.89 ± 0.07	1.52 ± 0.06	1.77 ± 0.09	2.84 ± 0.09	1.99 ± 0.12 *
22:1*n*-9, Erucic	0.17 ± 0.06	0.15 ± 0.06	0.25 ± 0.06	0.44 ± 0.04	0.43 ± 0.01	0.40 ± 0.04
24:1*n*-9, Nervonic	0.51 ± 0.10	0.59 ± 0.06	1.10 ± 0.05	1.27 ± 0.06	2.13 ± 0.06	1.71 ± 0.21
Total MUFA	21.10 ± 0.48	20.41 ± 0.33	26.61 ± 0.53	26.54 ± 0.61	32.55 ± 0.43	27.41 ± 0.99 *
18:3*n*-3, ALA	0.04 ± 0.04	ND	0.22 ± 0.07	ND	0.19 ± 0.02	ND
20:5*n*-3, EPA	0.11 ± 0.04	ND	0.10 ± 0.04	ND	0.18 ± 0.01	ND
22:5*n*-3, DPAn-3	0.66 ± 0.01	0.13 ± 0.13	0.87 ± 0.20	0.31 ± 0.26	0.91 ± 0.13	0.22 ± 0.15
22:6n-3, DHA	14.66 ± 0.39	12.01 ± 0.16 *	12.53 ± 0.50	9.81 ± 1.04	11.77 ± 0.25	9.95 ± 0.45
Total *n*-3 PUFA	15.46 ± 0.35	12.15 ± 0.15 *	13.72 ± 0.58	10.12 ± 1.10 *	13.06 ± 0.29	10.17 ± 0.37 *
18:2*n*-6, LA	0.94 ± 0.03	1.22 ± 0.03 *	0.89 ± 0.03	1.26 ± 0.02 *	0.75 ± 0.03	1.00 ± 0.03 *
20:2n-6, Eicosadienoic	0.17 ± 0.07	0.23 ± 0.09	0.22 ± 0.07	0.52 ± 0.02 *	0.40 ± 0.02	0.41 ± 0.03
20:3*n*-6, DGLA	0.47 ± 0.08	0.35 ± 0.08	0.53 ± 0.04	0.55 ± 0.03	0.59 ± 0.05	0.54 ± 0.05
20:4*n*-6, Arachidonic	10.32 ± 0.14	11.91 ± 0.1 *	8.76 ± 0.26	10.59 ± 0.13 *	7.94 ± 0.26	10.97 ± 0.42 *
22:4*n*-6, Adrenic	2.70 ± 0.06	4.01 ± 0.04 *	2.32 ± 0.08	3.64 ± 0.09 *	2.82 ± 0.14	4.06 ± 0.05 *
Total *n*-6 PUFA	14.61 ± 0.23	17.72 ± 0.11 *	12.72 ± 0.43	16.55 ± 0.24 *	12.50 ± 0.32	16.98 ± 0.42 *
*n*-3 PUFA: LC-PUFA	0.53 ± 0.01	0.43 ± 0.003 *	0.53 ± 0.01	0.39 ± 0.04 *	0.52 ± 0.01	0.39 ± 0.01 *
DHA:AA	1.42 ± 0.03	1.02 ± 0.004 *	1.43 ± 0.04	0.93 ± 0.10 *	1.49 ± 0.06	0.91 ± 0.02 *

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
