# Peer review of "Dietary n-6/n-3 Ratio Influences Brain Fatty Acid Composition in Adult Rats"

_nutrients, 2020, doi:10.3390/nu12061847_

Round 1

Reviewer 1 Report

The paper is well presented but there is a major point to be improve (see below), together with a few minor points to rectify as follows:

The following is a major point to be satisfied. The new version should express clearly the novelty of the experimental design and results achieved by authors.

-Page 5, Discussion subheading, 2nd paragraph.  Authors use reference 17 in 7 occasions throughout many sections of the manuscript, and I wonder if authors can compare and discuss what difference or novelty their work provide in relation to the reported ref. 17 (rats fed with different n-3 fatty acid contents, etc).  It sounds that they conducted a similar study, achieving similar results, decreasing the novelty of their own work. Please explain, rectify, and improve in the new version.  Perhaps, authors can also compare the results obtained with the used rat breed (Sprague-Dawley) in their research and compare it with the different breeds of rats in ref. 17, etc. etc. 

List of Minor points:

- Page 4, 4th line.  Total SFA level difference should be 7.8% as the real calculated value is 7.79%, but it is typed 7.7%.

-Page 5 Discussion, 3rd paragraph.  Authors wrote …..”This discrepancy is partially explained by the limited capacity of the brain tissue to store fatty acids [15”].   I didn´t find this statement in the cited ref. 15, a sound study which describes the metabolism of precursors (particularly ALA) to synthesize, primarily in the liver, different n-3 FA (fatty acids), which are then consumed in the brain, with precise techniques, etc.  Please rectify

In fact, it is well known that the brain has a very high content of FA for its proper physiology, showing both, capacity to store the required FA until use, and also a high capacity to consume them.  On the  contrary, brain capacity to de novo synthesize its needed FA is known to be very low and inefficient, as also indicated in ref. 15. 

Therefore, state that the brain has a limited capacity to store FA is not that correct.  If this were the case, how can authors explain all their FA data on the different brain tissues displayed in their Table 2.

I consider it is better to write that the ALA content in the analyzed brain tissue was very low due to for example a high conversion rate into other FA, which is known to occur mostly in the liver to them supply the brain with the needed levels of manufactured FA.  And other points to be added by authors.

Contradictory, further down in this paragraph authors suggest what I express here.  They wrote about ALA conversion in the liver, somehow contradicting such initial statement.

Please rectify these lines or remove that imprecise statement.

- Also in this paragraph, regarding the DHA levels in FD-fed rats, better type >360 as values difference were 366 fold. Please rectify it.

Reviewer 2 Report

It adds evidence to what has already been known for years about the increased deposit of essential fatty acids when they are provided in the diet. Only, I think that two aspects that are not well understood should be improved:
Table 1, has some aspects to improve, the abbreviation of grams is only. And in the ingredients section, they are placed in grams (g), but it does not say in relation to how much product. If you do it in relation to fatty acids (g fatty acid / kg diet).
Secondly, in relation to Table 2, it would be interesting to highlight the most relevant results, to make some figures of the distribution of at least the total of n3 and another with n6 in Prefrontal Cortex (PFC), hippocampus (HPC), and hypothalamus (HYP).
I think that by modifying those details, the article can be published

Reviewer 3 Report

The authors investigated in adult male Sprague-Dawley rats whether isocaloric diets (35% kcal from fat) differing in n-6 linoleic acid/n-3 alpha-linolenic acid ratio (high n-6/n-3 ratio ~600:1 or low n-6/n-3 ratio ~0.3:1) can impact fatty acid composition in prefrontal cortex, hippocampus, and hypothalamus, areas linked to various psychopathologies. They showed that the profiles of PUFA reflect the fatty acid composition of the diet and in the hypothalamus, but not in the hippocampus or prefrontal cortex, saturated fatty acids and monounsaturated fatty acids levels have changed. These changes on fatty acid remodeling in the adult rat brain can occur within months and could impact mental health.

The authors in experimental protocol did not use a standard control diet with a n-6/n-3 ratio of 5 according to recommendation or balanced diet in linoleic acid/alpha linolenic acid ratio, as showed in other papers Joffre, C et al. doi:10.1016/j.plefa.2016.09.003 or E. Zamberletti et al. DOI 10.1194/jlr.M068387.

In discussion on page 5, the authors write: “Moreover, DHA levels between the SD- and FD-fed rats, while statistically different, were of similar relative abundance.The authors do not explain the mechanism involved in this result. In fact, high dietary ALA inhibits the enzymatic activity involved in the metabolism of ALA, as described in other papers.

In conclusions the authors write: “that increasing dietary ALA can provide an alternate approach to EPA/DHA supplements for reshaping brain n-6/n-3 ratios.This claim cannot be supported by the data because a great conversion of DHA produced is not shown.

The authors do not report 22:5n6 fatty acid produced by n-3 PUFA deficient diet, as reported in other papers. The authors may verify their data?

Round 2

Reviewer 1 Report

I congratulate the authors for such interesting study and for soundly improving the manuscript following the reviewers comments.

There is just a minor spell error, Discussion, 3rd paragraph, line 6. It is typed . "There are at least two possible explanation", please rectify "explanations" should be in plural.

Author Response

There is just a minor spell error, Discussion, 3rd paragraph, line 6. It is typed . “There are at least two possible explanation”, please rectify “explanations” should be in plural.

Thank you for catching this error, this has been corrected in the revised Discussion, paragraph 3, line 6.

Reviewer 3 Report

I thank the authors for the adjustments made, but a variation in the data could be made to have an immediate statement of the results:

1) In fig1 c and d the authors indicated n3 PUFA and n6 PUFA respectively, perhaps it would be more appropriate to indicate the index n3UFA score (calculated as: sum EPAn3+DHAn3+DPAn3/sum all UFAs with = and > to 20C) in different brain regions and also to insert the DHA figure

2) In fig 1 the authors report Hip but in paper they use HPC

Author Response

1) In fig1 c and d the authors indicated n3 PUFA and n6 PUFA respectively, perhaps it would be more appropriate to indicate the index n3UFA score (calculated as: sum EPAn3+DHAn3+DPAn3/sum all UFAs with = and > to 20C) in different brain regions and also to insert the DHA figure.

Thank you for this suggestion. Panels e and f have been added to figure 1 indicating the n-3PUFA:LC-PUFA ratio and DHA:AA ratios.

2) In fig 1 the authors report Hip but in paper they use HPC.

Thank you for catching this error. This has ben corrected in the revised figure 1.